**Data Availability Statement:** All relevant data are within the manuscript; DOI: https://doi.org/10.5061/dryad.6gt43qg.

# The organization of Golgi in *Drosophila* bristles requires microtubule motor protein function and a properly organized microtubule array

**Anna Melkov[�], Raju Baskar[�], Rotem Shachal, Yehonathan Alcalay, Uri Abdu[iD]***

Department of Life Sciences, Ben-Gurion University of the Negev, Beer Sheva, Israel

☯ These authors contributed equally to this work.
* abdu@bgu.ac.il

## Abstract

In the present report, we used highly elongated *Drosophila* bristle cells to dissect the role of dynein heavy chain (Dhc64C) in Golgi organization. We demonstrated that whereas in the bristle "somal" region Golgi units are composed of cis-, medial, and trans-Golgi compartments ("complete Golgi"), the bristle shaft contains Golgi satellites that lack the trans-Golgi compartment (hereafter referred to as "incomplete Golgi") and which are static and localized at the base area. However, in *Dhc64C* mutants, the entire bristle shaft was filled with complete Golgi units containing ectopic trans-Golgi components. To further understand Golgi bristle organization, we tested the roles of microtubule (MT) polarity and the Dhc-opposing motor, kinesin heavy chain (Khc). For our surprise, we found that in *Khc* and *Ik2*$^{Dominant-negative\ (DN)}$ flies in which the polarized organization of MTs is affected, the bristle shaft was filled with complete Golgi, similarly to what is seen in *Dhc64C* flies. Thus, we demonstrated that MTs and the motor proteins Dhc and Khc are required for bristle Golgi organization. However, the fact that both *Dhc64C* and *Khc* flies showed similar Golgi defects calls for an additional work to elucidate the molecular mechanism describing why these factors are required for bristle Golgi organization.

## Introduction

The Golgi complex possesses a highly ordered and characteristic morphology. In most mammal cells, the organelle appears as stacks of flattened cisternae that are linked by tubules to give rise to continuous structures known as ribbons [1, 2]. In most cell types, the Golgi ribbon is typically located in the perinuclear region near the centrosome. The exact localization is dependent on the microtubule (MT) and actin cytoskeleton, together with active transport or anchoring by motor proteins, such as dynein and kinesin [3]. It was, moreover, shown that MTs are closely associated with the cis-Golgi compartment [4]. In contrast, in non-neuronal *Drosophila* tissues, the Golgi apparatus appears as stacks dispersed in the cell cytoplasm, juxtaposed to the endoplasmic reticulum exit site (ERES), and does not form the continuous tubular organelle seen in mammalian cells [5].

**Funding:** This work was supported by the Israel Science Foundation (ISF) (grant 278/16) to U.A.

**Competing interests:** The authors have declared that no competing interests exist.

Neurons, which are highly polarized cells, present a unique satellite secretory system that includes ER and Golgi outposts, preferentially localized to dendrites [6–8]. Although it is believed that the majority of protein synthesis, modification, and sorting takes place in the soma, it was shown that satellite Golgi at neuronal terminals is functional and may provide local membrane and protein supply [5, 6, 8, 9]. Moreover, it was demonstrated that dendritic Golgi outposts can serve as sites of *de novo* MT nucleation [10, 11], thus serving as microtubule-organizing centers (MTOCs). On the other hand, it was claimed that acentrosomal nucleation of MTs is only γ-tubulin-dependent and that Golgi outposts do not serve as sites for MT nucleation in neurons [12]. The appearance of distinct distal Golgi structures separated from the main Golgi apparatus and located near the nucleus mainly characterizes neuronal cells. However, discrete Golgi outposts are not only found in neurons. For instance, highly polarized mice astrocyte perivascular processes and end feet possess Golgi apparatus components [13], suggesting that the presence of Golgi outposts is not a unique feature of neurons, as was previously believed [14].

Similar to *Drosophila* dendrites, the bristle shaft contains acentrosomal stable and uniformly oriented MT arrays in which the minus-ends point distally (i.e., towards the growing edge of the bristle) [15]. Thus, the *Drosophila* bristle cell represents an excellent model for studying long-distance transport and the establishment of cellular polarity. The bristle shaft sprouts from the epithelial tissue for up to 450 μm, while the "soma", containing the polyploid nucleus, remains in the epithelial plane. During elongation, growth takes place at the tip area, which is the most dynamic part of the cell. Our lab and others have questioned the role of MT-dependent motors and asymmetrical MT organization in long-distance organelle transport [16–19]. Still, the organization of the Golgi in these unique polarized cells remains to be fully elucidated.

In the present study, we describe a previously uncharacterized Golgi apparatus in bristle cells. We also addressed the role of the MT array and its MT motor proteins, dynein and kinesin, in bristle Golgi organization. We found that in the somal region containing the nucleus, the Golgi contains cis-, medial, and trans-compartments juxtaposed to the ERES, while in the bristle shaft, the Golgi is composed of ERES, cis-, and medial compartments, yet completely lacks trans-Golgi components. We thus refer this complex as incomplete Golgi. These Golgi satellites were localized to the lower shaft area and appeared static. Relying on *dynein heavy chain 64C (Dhc64C)* and *kinesin heavy chain (Khc)* mutants to follow the contributions of these molecular motor to the positioning of Golgi satellites, we found that complete Golgi units containing trans-Golgi components were ectopically localized throughout the entire bristle shaft. Next, by affecting MT polarity through mutation of the *Ik2* gene, we found that complete Golgi units were localized ectopically to the entire bristle shaft; similar to what was seen in the *Dhc64C* and *Khc* mutants. Thus, our results demonstrate that organized MT polarity and the MT motor proteins Dhc64C and Khc are required for bristle Golgi organization.

## Materials and methods

### *Drosophila* stocks

Oregon-R was used as a wild-type control. The following mutant and transgenic flies were used: $Dhc64C^{4-19}$/TM6B, $Dhc64C^{8-1}$/TM6B [20], $FRT79D\ Dhc64C^{902}$/TM6B [21], $UAS\text{-}Ik2^{DN}$ [22], $UAS\text{-}LVA^{DN}$ [7], *Khc-IR* (Vienna Drosophila RNAi Center; V44337), *UAS-MnII-GFP*, *UAS-Gal-T-RFP* [7, 23], *UAS-GalNacT2-YFP/TagRFP* [23], and *UAS-GFP-Sten* [24]. Bristle expression was induced under the control of the *neur-Gal4* or *sca-Gal4* driver.

### Dissection and preparation of pupae for live imaging

Dissection of thorax tissue containing bristles for antibody staining and removal of the pupal case for live-imaging experiments was performed as previously described [15, 18].

## Scanning electron microscopy

Adult *Drosophila* were fixed and gradually dehydrated by immersion in increasing concentrations of ethanol (25%, 50%, 75%, and 2×100%, each for 10 min). The samples were then completely dehydrated using increasing concentrations of hexamethyldisilazane in alcohol (50%, 75%, and 2×100%, each for 2 h), air-dried overnight, placed on stubs, coated with gold, and examined with a scanning electron microscope (SEM; JEOL model JSM-5610LV).

## Bristle phalloidin and antibody staining

The procedures of tissue dissection used for fixation and staining were described previously [18, 25]. Confocal images were taken using an Olympus FV1000 laser scanning confocal microscope, appearing as Z-projections in a few optical frames that together covered the bristle cell. The primary antibodies used were mouse anti-acetylated tubulin monoclonal antibodies (1:250; Sigma), and anti-dGM130 (1:500; Abcam), anti-Sec16 antibodies [26]. Secondary antibodies included Cy3-conjugated goat anti-mouse (1:100; Jackson Immunoresearch) and Alexa Fluor 488-conjugated goat anti-rabbit (1:100; Molecular Probes) antibodies. For actin staining, we used Alexa Fluor 405-conjugated phalloidin (1:250; Molecular Probes).

## Organelle tracking and statistical analysis

To measure the area and density of MnII-GFP and GalNacT2-YFP-positive particles, confocal Z-stack images of maximal quality of bristle cells were obtained. A Z-stack projection of the bristle shaft was divided into two halves exactly in the middle of the shaft length, resulting in two sections of even length, with one part being close to the base (soma) and the other being distal to the base, referred to as the tip. In the case of a *dynein*-mutated background, the bristle appeared as two independent parts because of an extremely high signal at the tip area, relative to the base. Images were analyzed using ImageJ software, and the area of each particle was measured automatically using the "analyze particles" tool. The density of MnII and GalNacT2-positive particles was measured as the number of particles in each half of the bristle divided by the overall area of that half-bristle (i.e., the number of particles/area of the base or tip half). Statistical analysis of both parameters, i.e., particle area and density, was performed using spilt-plot ANOVA with the cell part area being considered as the within-plot treatment and the genotype being considered as the whole-plot treatment. Repeats (random factors) were nested within a genotype. Statistical analyses were performed using STATISTICA, version 10.

# Results

## Golgi organization in the *Drosophila* bristle cell

To study the organization of the Golgi in polarized cells, we used the highly elongated *Drosophila* bristle cell as a model system, together with a set of ERES and Golgi markers. To identify the ERES, we used either Stenosis (Sten) tagged with GFP [24] or antibody staining of Sec16 [26]. As a medial Golgi compartment marker, we followed α-mannosidase II tagged with EGFP (MnII-GFP), while RFP/GFP-tagged N-acetylgalactosaminyltransferase 2 (Gal-NacT2-TagRFP/GFP) and RFP-tagged Gal-T [23] served as a trans-Golgi marker. For visualization of the cis-Golgi compartment, we followed anti-dGM130 antibody staining [23].

   Examination of bristle cell Golgi organization revealed that the somal region contains Golgi stacks composed of cis- (Fig 1A' and 1A''), medial- (Fig 1A–1C, 1A'', 1B'' and 1C''), trans-Golgi compartments (Fig 1B and 1B') and ERES (Fig 1C' and 1C'') scattered throughout the cytoplasm. This type of Golgi is henceforth designated as a complete Golgi unit. Thus, similar to other *Drosophila* tissues [5, 27–29], the soma of the bristle cell contains Golgi mini-stacks

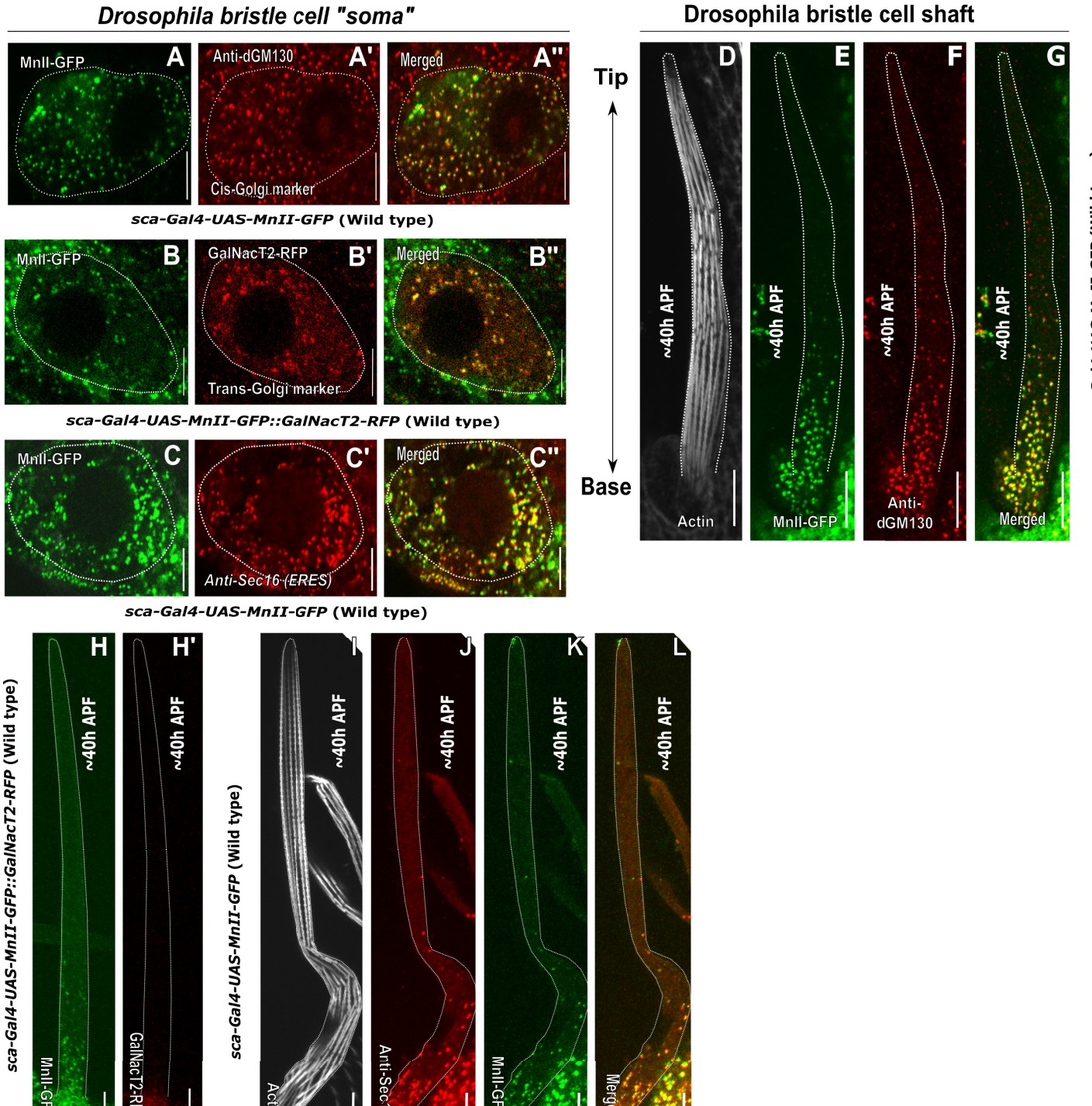

**Fig 1. Golgi organization in *Drosophila* bristles.** Confocal projections of representative WT "somal" and ~40 h APF bristle shaft areas. A-C"–Bristle "somal" region: A-A"–"Soma" of a WT bristle cell expressing *sca-Gal4::UAS-MnII-GFP* (a medial-Golgi marker) stained with anti-dGM130 antibodies (a cis-Golgi marker). A'–Anti-dGM130 antibody staining showing a cis-Golgi compartment localized throughout the bristle soma cytoplasm, A"–Merged image of green MnII-GFP and red anti-dGM130 antibody staining showing co-localization of medial- and cis-Golgi compartments. B-B"–Soma of a WT bristle cell expressing *sca-Gal4::UAS-MnII-GFP* (a medial-Golgi marker) and co-expressing GalNacT2-RFP (a trans-Golgi marker): B'–GalNacT2-RFP (a trans-Golgi marker) identifies trans-Golgi components localized throughout the bristle soma cytoplasm, B"–Merged image of green MnII-GFP and red GalNacT2-RFP showing co-localization of medial- and trans-Golgi compartments in the bristle cell soma. C-C'–Soma of a WT bristle cell expressing *sca-Gal4::UAS-MnII-GFP* (a medial-Golgi marker) stained with anti-Sec16-antibodies (to identify the ERES): C'–Anti-Sec16 antibody staining showing ERES localized throughout the bristle soma cytoplasm, C"–Merged image of green MnII-GFP and red

anti-Sec16 antibody staining showing co-localization of medial-Golgi and ERES components in the bristle cell soma. D-L–Bristle external extension; shaft (the cell compartment emanating from soma out from the epithelial tissue plane). D-G–Bristle shaft expressing *sca-Gal4::UAS-MnII-GFP* co-stained with anti-dGM130 antibodies and phalloidin: D–Gray phalloidin-UV staining of actin bundles in a MnII-GFP expressing bristle, used here to highlight the cell perimeter, E–Green MnII-GFP, a medial-Golgi marker, is localized to the lower shaft area close to the somal region, F–Red anti-dGM130 antibody staining (cis-Golgi marker), G–Merged image of green MnII-GFP and red anti-dGM130 antibody staining showing co-localization of medial- and cis-Golgi compartments in the bristle shaft. H-H'–Bristle shaft co-expressing *sca-Gal4::UAS-MnII-GFP* (a medial-Golgi marker) and GalNacT2-RFP (a trans-Golgi marker): H–Green MnII-GFP, a medial-Golgi marker, is localized to the lower shaft area close to the somal region, H'–Red GalNacT2-RFP, indicating a trans-Golgi domain, does not enter the bristle shaft. I-L–Bristle shaft expressing *sca-Gal4::UAS-MnII-GFP* co-stained with anti-Sec16 (ERES) antibodies and phalloidin: I–Phalloidin-UV staining of actin bundles in a MnII-GFP-expressing bristle, used here to highlight the cell perimeter, J–Red anti-Sec16 antibody staining localized to the lower shaft area close to the somal region, K–Green MnII-GFP localized to the lower shaft area close to the somal region, L–Merged image of green MnII-GFP and red anti-Sec-16-antibody staining showing co-localization of medial-Golgi and ERES components in the bristle shaft. APF-After prepupa formation. The scale bar represents 10 μm.

and does not create higher-order structures, such as the ring-shaped organization observed in the soma of larval neurons [23]. In *Drosophila* larval dendritic shafts, the Golgi exists as a discrete "single compartment Golgi" or as a "Golgi mini-stack" composed of more than one compartment [23]. Examination of Golgi organization within the bristle shaft using MnII-GFP as a medial-Golgi marker revealed that the shaft contains MnII-positive particles, these particles are called hereafter Golgi satellites, (Fig 1D–1G, 1H, 1K and 1L) and are concentrated in the lower part of the bristle, with almost no such Golgi satellites being seen in the distal to soma tip area. We tested for differences in the density of Golgi satellites and the relative sizes of particles in the lower part (i.e., the half of the shaft close to the soma, designated as the base) and the upper part (i.e., the half of the bristle shaft distal to the soma, designated as the tip) of the bristle shaft (see Materials and Methods for details of the quantification process used). Such analysis of MnII-GFP -positive particles revealed that the density of Golgi satellites (i.e., the number of particles per $\mu m^2$) at the base of the bristle shaft was significantly higher, as compared to the tip (For MnII-GFP, 0.18±0.05 and 0.04±0.06 particles/$\mu m^2$, respectively; P<0.001, Table 1; 5 pupa, 15 bristles). In addition, the average area of each particle of both MnII-GFP was significantly different between the base and tip areas, (For MnII-GFP, 0.43±0.80 and 0.17±0.20 $\mu m^2$, respectively; P<0.0015 Table 1; 5 pupa, 15 bristles)

We further found that these Golgi satellites to be composed of cis- (Fig 1D–1G) and medial-Golgi compartments (Fig 1E, 1H and 1K) juxtaposed to ERES (Fig 1I–1L) and completely lacking trans-Golgi domains (Fig 1H and 1H'). To confirm the complete absence of trans-Golgi components within the bristle shaft, as observed in GalNacT2-RFP-expressing bristles, we tested for the presence of an additional trans-Golgi marker, 1,4-galactosyltransferase (Gal-T)[23, 30]. We found no Gal-T-RFP in the wild type (WT) shaft (S1 Fig). Thus, our results show that whereas the Golgi apparatus of the bristle somatic region is composed of

**Table 1. Golgi outpost localization parameters in *Dhc64C* and *khc* mutant bristle shafts.**

| | Golgi outpost (MnII-GFP) localization parameters in *Drosophila* bristles | | | | | |
|---|---|---|---|---|---|---|
| Genotype | *Wild-type* | | *Dhc64C^8-1^/Dhc64C^4-19^* | | *Khc- RNAi* | |
| Bristle cell area | tip | base | tip | base | tip | base |
| No. of pupae | 5 | | 5 | | 5 | |
| No. of bristles | 15 | | 16 | | 15 | |
| Particle area, avg (μm²) | 0.17±0.20 | 0.43±0.80* | 0.43±0.84*^A | 0.32±0.52^A | 0.20±0.52^B | 0.40±0.19*^B |
| Density (particle/μm²) | 0.04±0.06 | 0.18±0.05* | 0.30±0.03^A | 0.43±0.16*^A | 0.12±0.13^A | 0.31±0.26*^A |

*—represents a significant difference of the cell part (tip/base) within each genotype; letters represent a significant difference in corresponding cell parts (tip/tip; base/base) between genotypes.

A–Significantly different from WT

B–significantly different from *Dhc64C* mutant. The values reflect mean±s.d. (for a detailed description of the statistical analysis performed, see Materials and Methods).

complete Golgi units (i.e, including cis-, medial- and trans-Golgi components), the bristle shaft contained Golgi satellites completely lacking trans-Golgi components.

## *Dhc64C* is required for Golgi organization in the bristle shaft

Given that in the *Drosophila* bristle, a stable MT array is organized with minus-end MT fibers pointing away from the soma [15] and with the notion that MTs and their motor proteins are required for Golgi organization [31–34], we studied the role of the minus-end MT motor protein dynein in bristle Golgi organization. First, we analyzed the role of *Dhc64C* using the "slow dynein" hypomorphic mutant allele *Dhc64C*[8-1] [17, 20]. We found that in these mutants, Golgi satellites were no longer concentrated at the base of the bristle shaft as in the WT (Fig 1E–1G) but were instead found throughout the entire bristle shaft (Fig 2A–2L). These ectopically local- ized Golgi units contained cis- (Fig 2C, 2D, 2G and 2H), medial- (Fig 2B, 2D, 2J and 2L), and trans-Golgi compartments (Fig 2F and 2H) and ERES (Fig 2K and 2L), demonstrating that in *Dhc64C* mutants, the bristle shaft contains complete Golgi units rather than discrete Golgi sat- ellites. Staining of MnII-GFP-expressing *Dhc64C* mutants with anti-dGM130 antibodies (a cis-Golgi marker) (Fig 2C) revealed co-localization of cis- and medial-Golgi markers (Fig 2D). Likewise, co-localization of GalNacT2-YFP (a trans-Golgi marker) expressed in *Dhc64C* mutants with anti-dGM130 antibody staining (Fig 2F–2H) reflected co-localization of the trans- and cis-markers (Fig 2H). Also, an additional trans-Golgi marker, 1,4-galactosyltrans- ferase (Gal-T), was found throughout *Dhc64C* mutant bristle shafts (S1 Fig). We thus con- cluded that all three compartments are co-localized. Additionally, we showed that MnII-GFP was co-localized with ERES (Fig 2L). Thus, the Golgi particles found within the *Dhc64C* mutant bristle are complete Golgi units. Quantitative analysis revealed that the density of MnII-GFP-positive particles was slightly higher at the base than at the tip (For MnII-GFP, 0.43 ±0.16 and 0.30±0.03 particles/$\mu m^2$, respectively; P<0.05; Table 1; 5 pupa, 16 bristles). For Gal- NacT2-YFP, 0.59±0.06 and 0.24±0.02 particles/$\mu m^2$, respectively; P<0.05; Table 2; 5 pupa, 16 bristles. In addition, the density of MnII-GFP-positive particles was dramatically increased in both the tip and base, as compared to the WT (P<0.002 for both tip and base; Table 1; 5 pupa, 16 bristles). Such analysis was not possible for GalNacT2-YFP due to its absence from the WT bristle shaft. Also, the average area of both MnII-GFP- and GalNacT2-YFP-positive particles was significantly larger at the tip of the bristle shaft than at the base (For MnII-GFP, 0.46±0.63 and 0.29±0.39 $\mu m^2$, respectively; P<0.001; Table 1; 5 pupa, 16 bristles). For GalNacT2-YFP, 0.53±0.03 and 0.26±0.20 $\mu m^2$, respectively; P<0.003; Table 2; 5 pupa, 16 bristles. Moreover, these areas of MnII-GFP were significantly (P<0.001) different from what was seen in the cor- responding cell parts in WT flies (Table 1; this analysis was not possible for GalNacT2-YFP since the WT bristle did not contain any trans-compartments in the shaft to analyze, Table 2). The defects in Golgi organization in the *Dhc64C* mutants bristle shaft led us to check whether the somal region of the bristle was affected. The closer examination revealed that in *Dhc64C* mutants, the Golgi apparatus is scattered throughout the somal region (S2 Fig), as in WT.

## Lava-lamp is not required for Golgi organization in the bristle shaft

Previously it was shown that in *Drosophila* embryo cellularization, Golgi movement depends on cytoplasmic dynein [35]. Moreover, it was shown that the Golgin (Golgi-associated protein) Lava-lamp (Lva) mediates such dynein-based Golgi movements [35, 36]. Later, it was shown that Lva is also required for the dynamic profile of dendritic Golgi satellites [37]. Thus, we ana- lyzed the role of Lva in bristle Golgi organization by disrupting Lva activity via over-expression of the Lva-dominant negative (*Lva*[DN]) protein [7]. We found that expression of *Lva*[DN] in bris- tles using *neur-Gal4* but not *sca-Gal4* affected bristle morphology, with the upper part of the

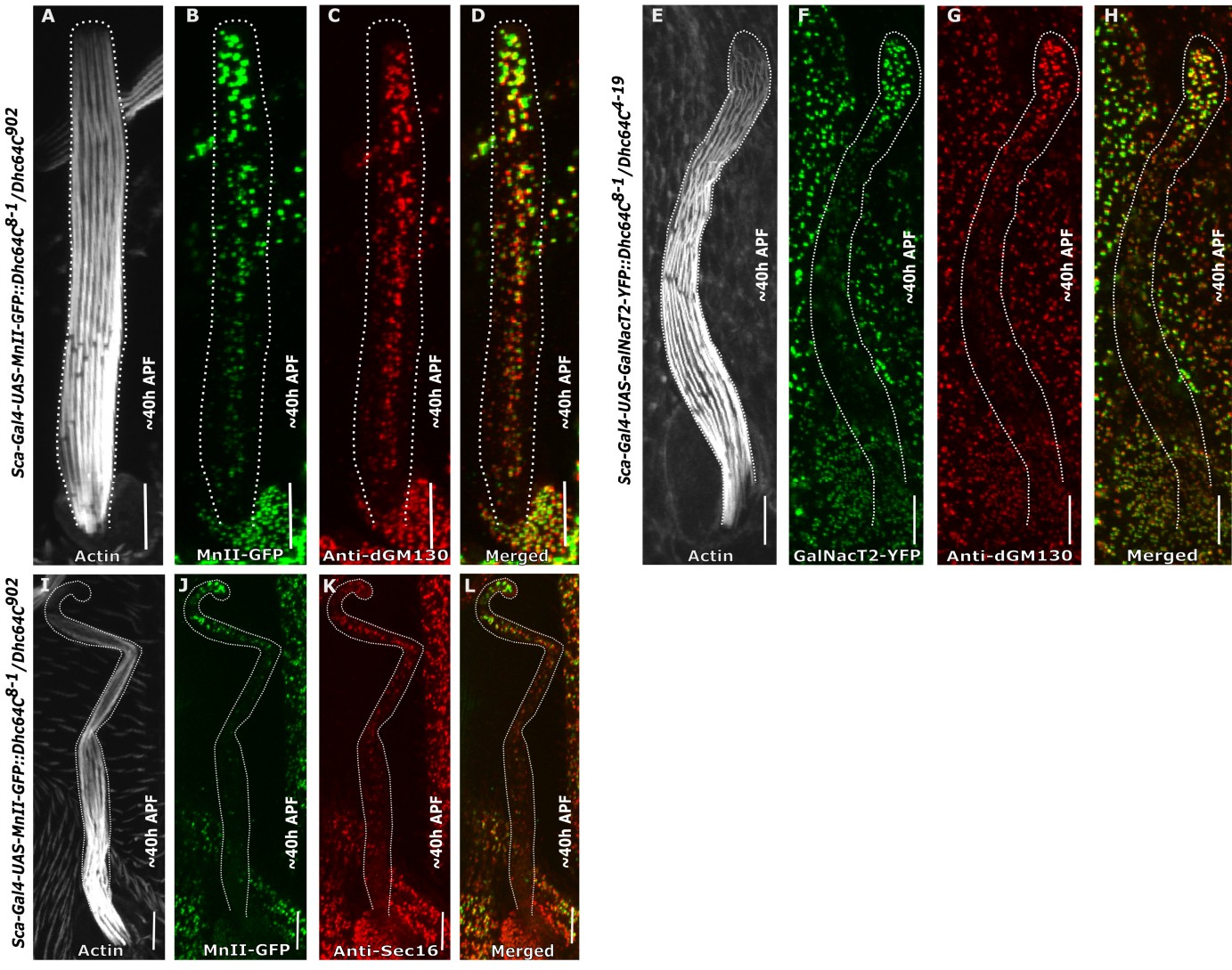

**Fig 2. Golgi organization in *Dhc64C* mutant bristle cell shaft.** Confocal projections of *Dynein heavy chain*-mutated background bristle shafts from ~40 h APF. A-D–*Dhc64C$^{8-1}$/Dhc64C$^{902}$* trans-heterozygote bristle shaft expressing *sca-Gal4*::*MnII-GFP* stained with anti-dGM130 (cis-Golgi marker) antibodies and phalloidin: A–Gray phalloidin-UV staining of actin bundles in a MnII-GFP-expressing bristle, used here to highlight the cell perimeter: B–Green MnII-GFP, a medial-Golgi marker, is localized to the entire bristle shaft area, C–Red anti-dGM130 antibody staining (cis-Golgi marker), D–Merged image of green MnII-GFP and red anti-dGM130 antibody staining showing co-localization of medial- and cis-Golgi compartments in the bristle shaft. E–H–*Dhc64C$^{8-1}$/Dhc64C$^{4-19}$* trans-heterozygote bristle shaft expressing *sca-Gal4*::*GalNacT2-YFP* (trans-Golgi marker) co-stained with anti-dGM130 antibodies and phalloidin: E–Gray phalloidin-UV staining of actin bundles in a MnII-GFP-expressing bristle, used here to highlight the cell perimeter: F–Green GalNacT2-YFP trans-Golgi marker is ectopically localized to the entire bristle shaft area, G–Red anti-dGM130 antibody staining (cis-Golgi marker), H–Merged image of green GalNacT2-YFP and red anti-dGM130 antibody staining showing co-localization of trans- and cis-Golgi compartments in the mutant bristle shaft. I-L–*Dhc64C$^{8-1}$/Dhc64C$^{902}$* trans-heterozygote bristle shaft expressing *sca-Gal4*::*MnII-GFP*, co-stained with anti-Sec16 (ERES) antibodies and phalloidin: I–Gray phalloidin-UV staining of actin bundles in a MnII-GFP-expressing bristle, used here to highlight the cell perimeter: J—Green MnII-GFP, a medial-Golgi marker, is localized to the entire bristle shaft area, K–Red anti-Sec16 antibody staining localized to the entire shaft area, L–Merged image of green MnII-GFP and red anti-Sec-16-antibody staining showing co-localization of medial-Golgi and ERES components in the mutant bristle shaft. APF—After prepupa formation. Scale bar, 10 μm.

bristle being twisted (90% of the bristle in 10 adults; Fig 3B and 3B'). On the other hand, using the same *neur-Gal4*, no defects in Golgi satellite organization were detected at the bristle shaft (Fig 3G–3J, S3 Fig) when compared to WT (Fig 3C–3F, S3 Fig), and no defects were found in the bristle somal region (S4 Fig).

**Table 2. Golgi outpost localization parameters in *Dhc64C* and *khc* mutant bristle shafts.**

| Golgi outpost (GalNacT2-YFP) localization parameters in *Drosophila* bristles | | | | | |
|---|---|---|---|---|---|
| Genotype | *Wild-type* | | *Dhc64C^8-1^/Dhc64C^A-19^* | | *Khc- RNAi* |
| Bristle cell area | tip | base | tip | base | tip | base |
| No. of pupae | 5 | | 5 | | 5 | |
| No. of bristles | 15 | | 16 | | 15 | |
| Particle area, avg (μm²) | N/A | N/A | 0.53±0.03* | 0.26±0.20 | 0.23±0.18 ^A | 0.33±0.36*^A |
| Density (particle/μm²) | N/A | N/A | 0.24±0.02 | 0.59±0.06* | 0.23±0.23 | 0.47±0.41* |

*—represents a significant difference of the cell part (tip/base) within each genotype; letters represent a significant difference in corresponding cell parts (tip/tip; base/base) between genotypes.

A–Significantly different from *Dhc64C* mutant. The values reflect mean±s.d. (for a detailed description of the statistical analysis performed, see Materials and Methods).

N/A- Not applicable.

## Kinesin heavy chain is required for Golgi organization in the bristle shaft

Next, we examined the role of the plus-end-directed MT motor kinesin, and more specifically, the conventional kinesin-41, kinesin heavy chain (Khc), in bristle Golgi organization. Although dynein is believed to be the primary motor in Golgi positioning, it was shown in vertebrate cells that kinesin-light chain is present in Golgi membranes [38]. Still, the role of kinesin in Golgi trafficking is not well understood. Accordingly, we down-regulated the *Khc* transcript specifically in the bristle using *RNAi* (for simplicity, these strains are henceforth termed *Khc* mutants). To our surprise, we found that similar to *Dhc64C* mutant flies, bristles in *Khc* mutants also contained complete Golgi units located throughout the entire bristle shaft (Fig 4) composed of co-localized cis- (Fig 4C and 4D), medial- (Fig 4B, 4D, 4J and 4L), and trans-Golgi components (Fig 4F and 4H) and ERES (Fig 4K and 4L). However, the distribution pattern of the Golgi units in the *Khc* mutant bristle shafts was different from that in the *Dhc64C* mutants. Similar to WT and *Dhc64C* mutant flies (Tables 1 and 2), the density of MnII-GFP was significantly higher at the base of the bristle than at the tip (For MnII-GFP, 0.31±0.26 particle/μm² and 0.12±0.13 particle/μm², respectively (P<0.001); Table 1; 5 pupa, 15 bristles). Moreover, the density of MnII-positive particles at both the tip and base was significantly higher than recorded for the WT (Table 1, P<0.0001) but similar to *Dhc64C*. However, in contrast to *Dhc64C* mutants, the average area of the MnII-GFP and GalNacT2-positive particles was significantly larger at the base of the bristle shaft than at the tip (For MnII-GFP, 0.40 ±0.19 μm² and 0.20±0.52 μm², respectively. For GalNacT2-YFP, 0.33±0.36 μm² and 0.23 ±0.18 μm², respectively, Table 2, 5 pupa, 15 bristles). Also, in the *Khc* mutant bristle somal region, no obvious defects in Golgi organization were detected (S5 Fig).

## Mutation of *Ik2* affects Golgi organization in the bristle shaft

We previously demonstrated that in the bristle, stable MTs are organized in an asymmetrical manner, with the minus-end pointing outwards [15]. Moreover, we found that the Spn-F/Ik2 complex is required for bristle MT polarity [39]. Thus, to test whether MT polarity is essential for bristle Golgi organization, we studied Golgi organization in flies expressing *Ik2^dominant-negative^* (*Ik2^DN^*) specifically in the bristle. *Ik2^DN^* is a defective kinase in which lysine at position 41 was changed to alanine (*Ik2^K41A^*). We and other have shown that expression of *Ik2^DN^* strongly affects bristle morphology and MT organization [15, 40, 41]. First, we found that expression with *neur-Gal4* but not *sca-Gal4* generated typical *Ik2* bristle defects. Thus, we used the *neur-Gal4* for further examinations. We found that similar to *Dhc64C* and *Khc* mutant flies, in *Ik2^DN^* mutants, the bristle contained complete Golgi units localized throughout the entire bristle shaft (Fig 5A–

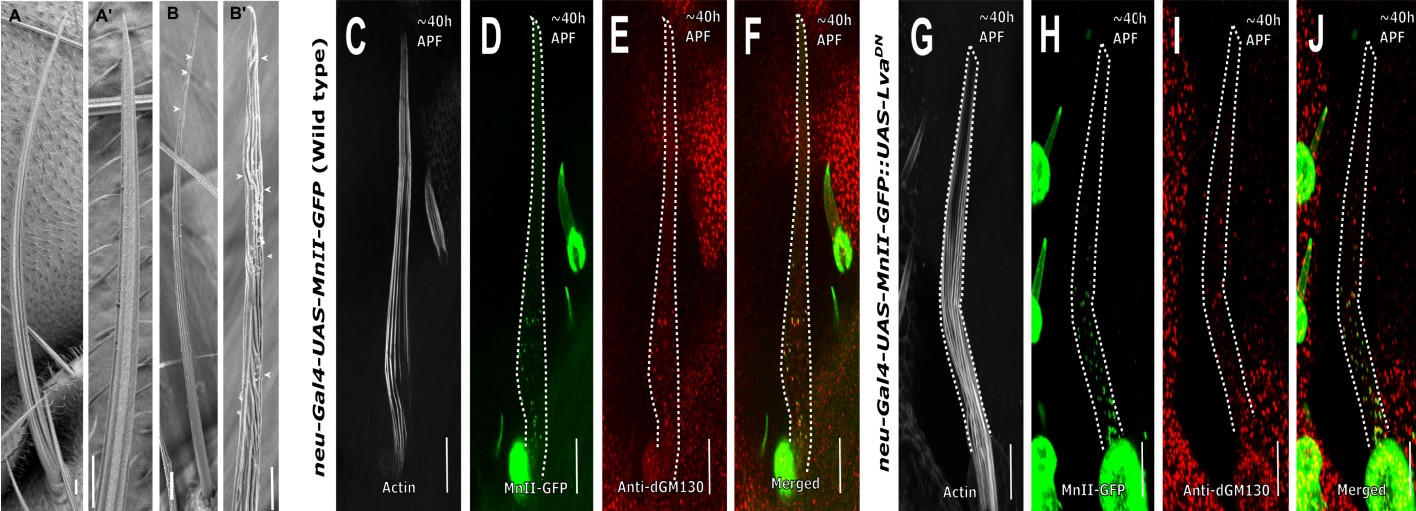

**Fig 3. Lava-lamp is not required for Golgi organization in the bristle shaft.** Scanning electron microscope images (A-B') of adult bristles from A–Wild type, A'–higher magnification image of the tip area showing characteristic parallel grooved surface morphology; B–*neur-Gal4::UAS-LVA*$^{DN}$ fly bristle, B'—higher magnification image of the tip area of *neur-Gal4::UAS-LVA*$^{DN}$ showing disrupted characteristic parallel grooved surface morphology highlighted by arrowheads. C-F- Confocal projections of *neur-Gal4::UAS-MnII-GFP* (Wild type) bristle shafts from ~40 h APF stained with anti-dGM130 (cis-Golgi marker) antibodies and phalloidin: C–Gray phalloidin-UV staining of actin bundles in a MnII-GFP-expressing bristle, used here to highlight the cell perimeter: D–Green MnII-GFP, a medial-Golgi marker, is localized to the base of bristle shaft area, E–Red anti-dGM130 antibody staining (cis-Golgi marker), F–Merged image of green MnII-GFP and red anti-dGM130 antibody staining showing co-localization of medial- and cis-Golgi compartments in the bristle shaft. G-J- Confocal projections of *Lava-lamp*$^{DN}$ background bristle shafts. G-J- *neur-Gal4::UAS-Lava-lamp*$^{DN}$ bristle shafts from ~40 h APF co-expressing *UAS-MnII-GFP* and stained with anti-dGM130 (cis-Golgi marker) antibodies and phalloidin: G–Gray phalloidin-UV staining of actin bundles in a MnII-GFP-expressing bristle, used here to highlight the cell perimeter: H–Green MnII-GFP, a medial-Golgi marker, is localized to the entire bristle shaft area, I–Red anti-dGM130 antibody staining (cis-Golgi marker), J–Merged image of green MnII-GFP and red anti-dGM130 antibody staining showing co-localization of medial- and cis-Golgi compartments in the bristle shaft. APF—After prepupa formation. Scale bar, 10 μm.

5L (and co-localized cis- (Fig 5C, 5D, 5G and 5H), medial- (Fig 5B, 5D, 5J and 5L) and trans-Golgi components (Fig 5F and 5H) and ERES (Fig 5K and 5L). Since we used different Gal4s for Golgi expression in *Dhc64C* and *Khc-RNAi* (*sca-Gal4*), in comparison to *Ik2*$^{DN}$ expression (*neur-Gal4*), we were not able to statistically compare Golgi parameter (area and density) between them. Instead, we compared the *neur-Gal4::UAS-MnII-GFP* and *GalNacT2*-positive particle densities in *Ik2*$^{DN}$ mutant bristle shafts versus their Gal4-UAS genetic background (*neur-Gal4::UAS-MnII-GFP* and *GalNacT2-YFP*).

First, we noticed that there was no difference in MnII-GFP-positive particle average area between the tip (0.32±0.23 μm$^2$, Table 3) and base (0.32±0.28 μm$^2$, Table 3) in the *Ik2* mutant bristle shaft. The same pattern was also detected in GalNacT2-positive particle average area (tip: 0.24±0.04 μm$^2$; base: 0.24±0.09 μm$^2$, Table 3) from the *Ik2* mutant. Interestingly, MnII-GFP-positive particles density was higher at the tip (0.26±0.10 particle/μm$^2$, P<0.001; 5 pupa, 18 bristles, Table 3) but was similar at the base to what was noted in the WT (Table 3). In case of GalNacT2-YFP, this analysis was impossible because of its absence (Table 3). Moreover, in *Ik2*$^{DN}$ mutants no defects in Golgi organization at the somal region of the bristle could be detected (S6 Fig).

## Differential effects on Golgi organization among *Dhc*, *Khc* and *Ik2* mutants

To understand the molecular mechanism by which *Dhc64C*, *Khc* and *Ik2*$^{DN}$ affect bristle Golgi organization; we followed the timing of Golgi organization in these mutants during bristle development. Since we used a different Gal4 for *Ik2*$^{DN}$ (*neur-Gal4*) than used with the *Dhc64C* and *Khc* mutants (*sca-Gal4*), we could only compare between *Dhc64C* and *Khc* mutants, with *Ik2*$^{DN}$ being compared to the control. We found that at early stages of WT bristle development,

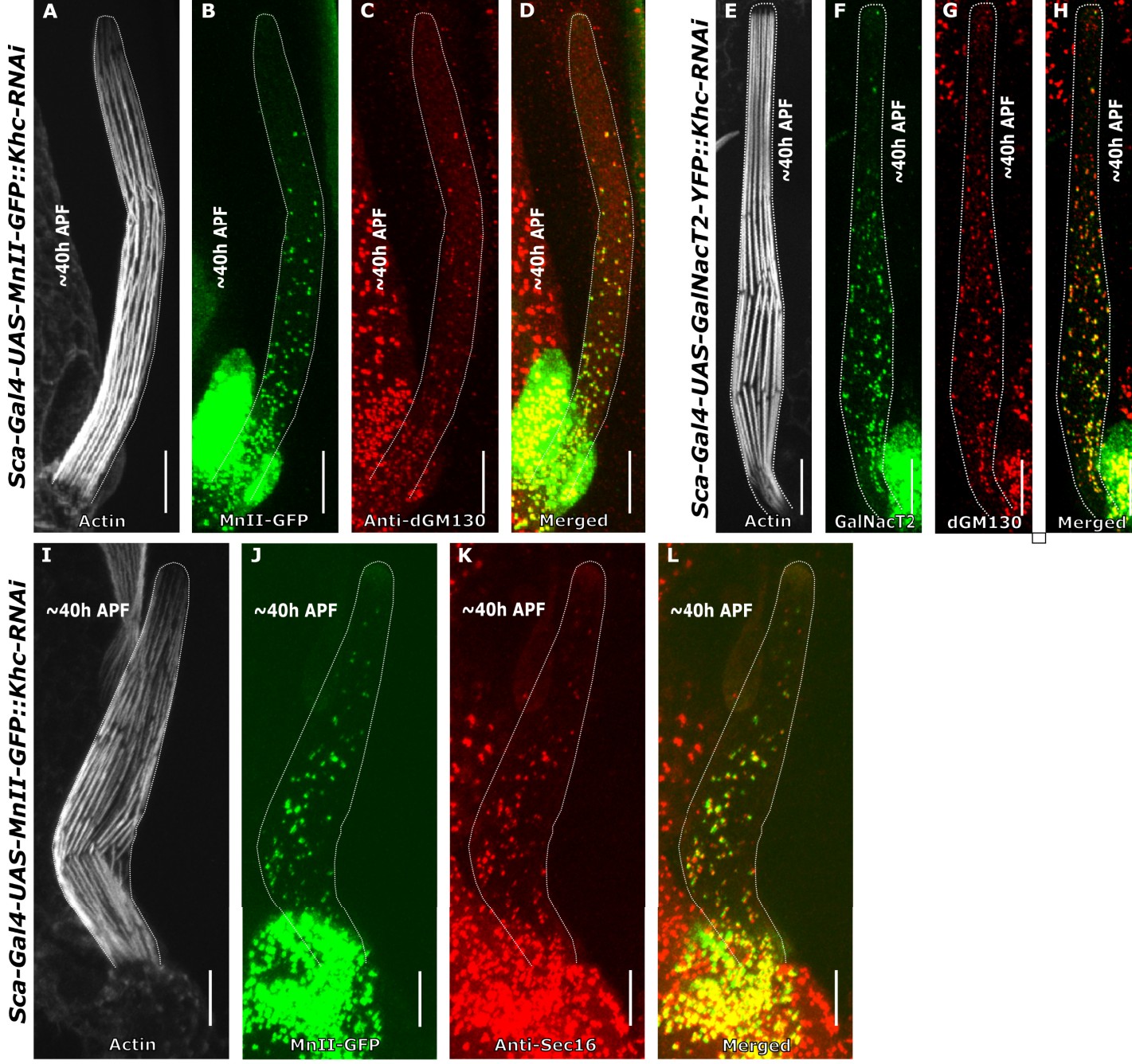

**Fig 4. Golgi organization in *kinesin heavy chain*-mutated bristles.** Confocal projections of *kinesin heavy chain*-mutated background bristle shafts from ~40 h APF. **A-D**–*Khc-RNAi* bristle shaft expressing *sca-Gal4*::*UAS*-MnII-GFP co-stained with anti-dGM130 (cis-Golgi marker) antibodies and phalloidin: **A**–Gray phalloidin-UV staining of actin bundles in a MnII-GFP-expressing bristle, used here to highlight the cell perimeter, **B**–Green MnII-GFP, a medial-Golgi marker, is localized to the entire bristle shaft area, **C**–Red anti-dGM130 antibody staining (cis-Golgi marker), **D**–Merged image of green MnII-GFP and red anti-dGM130 showing co-localization of medial- and cis-Golgi compartments in the bristle shaft. **E-H**–*Khc-RNAi* bristle shaft expressing *sca-Gal4*::*UAS*- GalNacT2-YFP (trans-Golgi marker) co-stained with anti-dGM130 antibodies and phalloidin: **E**–Gray phalloidin-UV staining of actin bundles in a GalNacT2-YFP-expressing bristle, used here to highlight the cell perimeter, **F**–Green GalNacT2-YFP, a trans-Golgi marker, is ectopically localized to the entire bristle shaft area, **G**–Red anti-dGM130 antibody staining (cis-Golgi marker), **H**–Merged image of green GalNacT2-YFP and red anti-dGM130 showing co-localization of trans- and cis-Golgi compartments in a mutated bristle shaft. **I-L**–*Khc-RNAi* bristle shaft expressing *sca-Gal4*::*UAS*-MnII-GFP co-stained with anti-Sec16 (ERES) antibodies and phalloidin: **I**–Gray phalloidin-UV staining of actin bundles in a MnII-GFP-expressing bristle, used here to highlight the cell perimeter, **J**—green MnII-GFP, a medial-Golgi marker, is localized to the entire bristle shaft area, **K**–Red anti-Sec16 antibody staining localized to the entire shaft area, **L**–merged image of green MnII-GFP and red anti-Sec-16-antibody showing co-localization of medial- and ERES components in a mutated bristle shaft. APF—After prepupa formation. Scale bar, 10 μm.

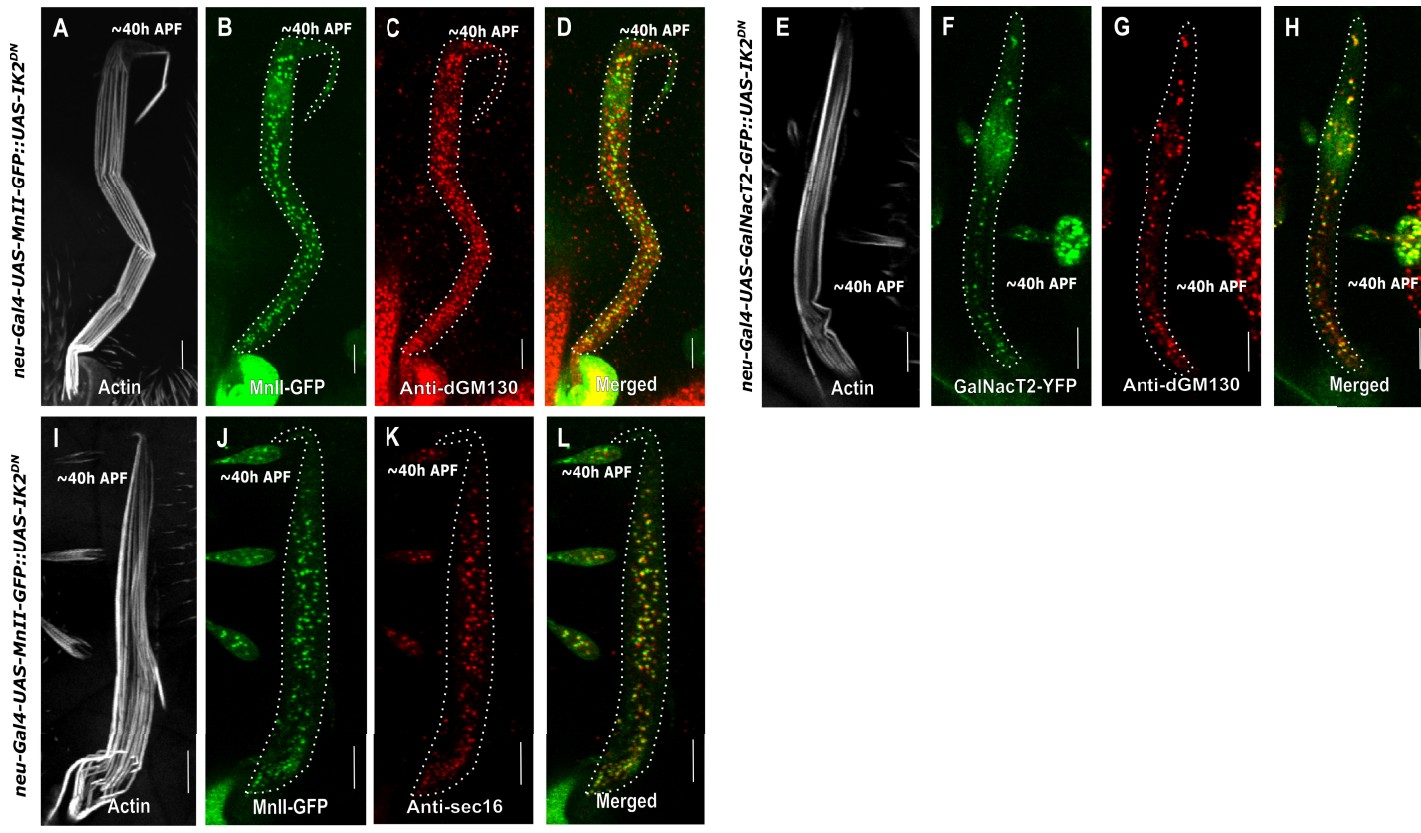

**Fig 5. Confocal projections of *Ik2^DN* background bristle shafts from ~40 h APF.** A–D–*neur-Gal4::UAS-Ik2^DN* bristle shaft co-expressing *UAS-MnII-GFP* and stained with anti-dGM130 (cis-Golgi marker) antibodies and phalloidin: A–Gray phalloidin-UV staining of actin bundles in a MnII-GFP-expressing bristle, used here to highlight the cell perimeter: B–Green MnII-GFP, a medial-Golgi marker, is localized to the entire bristle shaft area, C–Red anti-dGM130 antibody staining (cis-Golgi marker), D–Merged image of green MnII-GFP and red anti-dGM130 antibody staining showing co-localization of medial- and cis-Golgi compartments in the bristle shaft. E–H–*neur-Gal4::UAS-Ik2^DN* bristle shaft co-expressing *UAS-GalNacT2-YFP* (trans-Golgi marker) stained with anti-dGM130 antibodies and phalloidin: E–Gray phalloidin-UV staining of actin bundles in a MnII-GFP-expressing bristle, used here to highlight the cell perimeter: F–Green GalNacT2-YFP, a trans-Golgi marker, is ectopically localized to the entire bristle shaft area, G–Red anti-dGM130 antibody staining (cis-Golgi marker), H–Merged image of green GalNacT2-YFP and red anti-dGM130 antibody staining showing co-localization of trans- and cis-Golgi compartments in the mutant bristle shaft. I-L–*neur-Gal4::UAS-Ik2^DN* bristle shaft co-expressing *UAS-MnII-GFP* stained with anti-Sec16 (ERES) antibodies and phalloidin: I–Gray phalloidin-UV staining of actin bundles in a MnII-GFP-expressing bristle, used here to highlight the cell perimeter: J—Green MnII-GFP, a medial-Golgi marker, is localized to the entire bristle shaft area, K–Red anti-Sec16 antibody staining is localized to the entire shaft area, L–Merged image of green MnII-GFP and red anti-Sec-16-antibody showing co-localization of medial-Golgi and ERES components in the mutant bristle shaft. APF—After prepupa formation. Scale bar, 10 μm.

i.e. until 30–35 h APF (After prepupa formation), no Golgi satellites were detected in the bristle shaft in the WT (Fig 6A–6B). However, in both the *Dhc64C* (Fig 6D and 6E) and *Khc* (Fig 6G and 6H) mutants, even in the early stages of elongation, e.g., ~32 h APF, MnII-GFP positive particles could be detected throughout the bristle shaft. As mentioned above, later in the development of the *Dhc64C* and *Khc* mutants, the Golgi were spread all over the bristle shaft (Fig 6F and 6I). In *Ik2^DN*, the timing of MnII-GFP entry into the bristle shaft was at 38 h APF in bristle development (Fig 7F).

## Discussion

### Golgi organization in *Drosophila* bristles

In different *Drosophila* tissues, the Golgi shows different organizational patterns, reflecting differences in cell type. In most *Drosophila* cells, the Golgi is organized as discontinuous stacks composed of ERES and cis-, medial-, and trans-Golgi compartments [5]. During

**Table 3. Golgi outpost localization parameters in Ik2$^{DN}$ bristles.**

| | Golgi outpost localization parameters in Drosophila bristles | | | | | | | |
|---|---|---|---|---|---|---|---|---|
| Genotype | Wild-type | | Ik2$^{DN}$ | | Wild-type | | Ik2$^{DN}$ | |
| Golgi compartments | Medial- compartment (mnII) | | Medial-compartment (mnII) | | Trans-compartment (GalNacT2) | | Trans-compartment (GalNacT2) | |
| Bristle cell area | Tip | Base | Tip | Base | Tip | Base | Tip | Base |
| No. of pupae | 5 | | 5 | | 5 | | 5 | |
| No. of bristles | 15 | | 18 | | 18 | | 18 | |
| Particle area avg (µm$^2$) | 0.17±0.20 | 0.43±0.80* | 0.32±0.23$^A$ | 0.32±0.28 | N/A | N/A | 0.24±0.04 | 0.24±0.09 |
| Density (particle/µm$^2$) | 0.04±0.06 | 0.18±0.05* | 0.26±0.10$^A$ | 0.20±0.32$^A$ | N/A | N/A | 0.30±0.03 | 0.23±0.07 |

*—represents a significant difference of the cell part (tip/base) within each genotype; letters represent a significant difference in corresponding cell parts (tip/tip; base/base) between genotypes.

A–Significantly different from WT. The values reflect mean±s.d. (for a detailed description of the statistical analysis performed, see Materials and Methods).

N/A- Not applicable.

spermatogenesis, the Golgi-based acroblast has a ring shape [42, 43]. In *Drosophila* larval neurons, the Golgi is differently organized. Here, the soma contains a high-ordered ring-shaped complete Golgi structure. In addition, similar to mammals, the dendrites contain either a discrete "single compartment Golgi" or "Golgi mini-stacks" composed of more than one Golgi compartment [23]. In our study, we showed that bristle cells contain complete Golgi stacks at the soma and incomplete Golgi composed of ERES, and cis-, and medial-Golgi compartments at the shaft. We show for the first time specifically in *Drosophila* that Golgi satellites can be

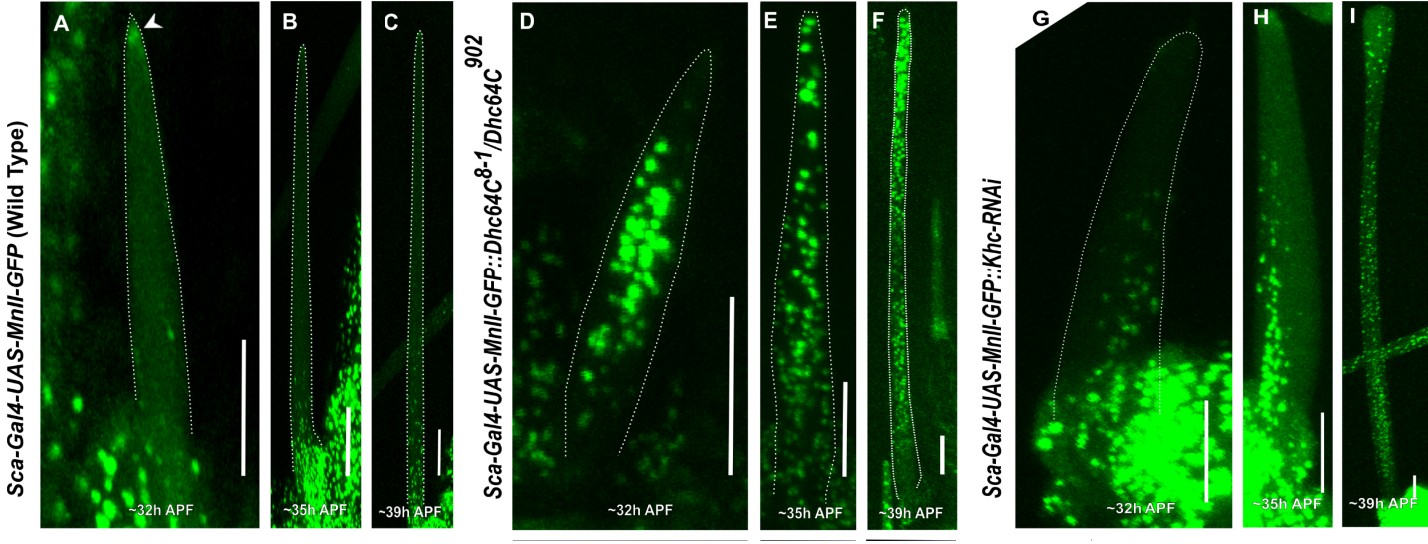

**Fig 6. Comparing developmental stages as a function of Golgi outpost localization.** Confocal projections of live WT developing bristle shafts expressing *sca-Gal4::MnII-GFP* (a medial-Golgi marker) at different elongation stages (A-C). A–~32 h APF, the developing bristle shaft shows no MnII-positive particles at this early stage of bristle development. Arrow points to the accumulation of MnII-GFP protein at the bristle tip, B–~35h APF, the developing bristle shaft shows no Golgi outposts localized to the bristle shaft. C–~39 h APF, the developing bristle shaft shows Golgi outposts beginning to accumulate at the base of the bristle shaft. D-F–*Dynein heavy chain* (*Dhc64C$^{8-1}$/Dhc64C$^{902}$* trans-heterozygotes)-mutated developing bristle shafts expressing *sca-Gal4::MnII-GFP*: D–~32 h APF, the developing *Dhc64C$^{8-1}$/Dhc64C$^{902}$* background bristle shaft shows MnII-positive particles filling the entire shaft area at this early stage of bristle development, E–~35 h APF, Golgi outposts are found throughout entire bristle shaft. The same is true for the following developmental stage of bristle shaft elongation: F– 39 h APF. G-I–developing bristle shafts expressing both *sca-Gal4::MnII-GFP* and *UAS-Khc-RNAi;* images show the corresponding developmental staging of *Khc*-depleted bristles, depicting the early appearance of Golgi outposts, G– 32 h APF, H– 35 h APF, I–At an advanced developmental stage (~39h APF), Golgi outposts localize evenly throughout the entire bristle shaft. APF—After prepupa formation. Scale bar, 10 µm.

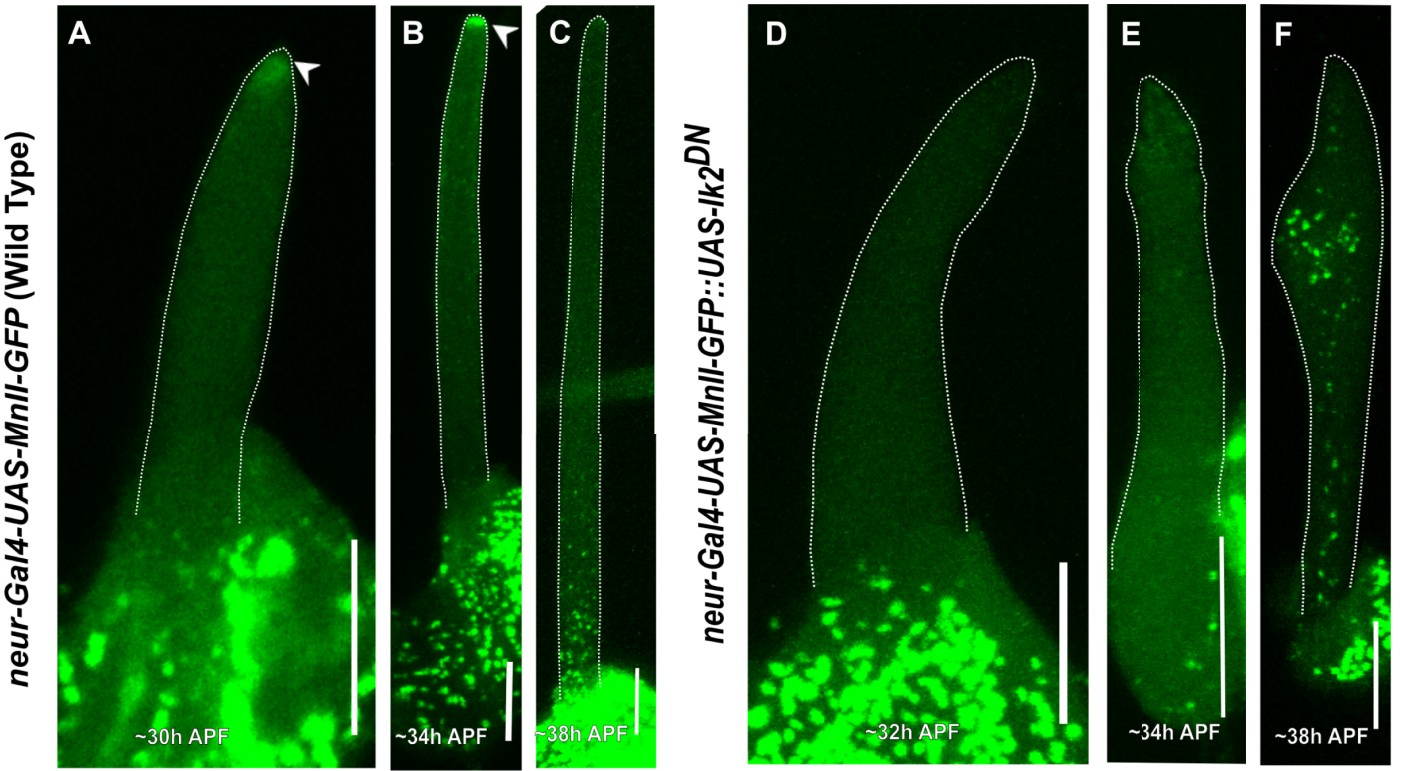

**Fig 7. Comparing developmental stages as a function of Golgi outpost localization in *Ik2^{DN}* bristles.** Confocal projections of live WT developing bristle shafts expressing *MnII-GFP* by *neur-Gal4* (a medial-Golgi marker) at different elongation stages (**A-C**). **A**–~32 h APF, the developing bristle shaft shows no MnII-positive particles at this early stage of bristle development. Arrow points to the accumulation of MnII-GFP protein at the bristle tip, **B**–~34h APF, the developing bristle shaft shows no Golgi outposts localized to the bristle shaft. Arrow points to the accumulation of MnII-GFP protein at the bristle tip, **C**–~38 h APF, the developing bristle shaft shows Golgi outposts beginning to accumulate at the base of the bristle shaft. **D-F**—*neur-Gal4::UAS-Ik2^{DN}*-developing bristles co-expressing *UAS-MnII-GFP*, **D**—~32 h APF, the developing bristle shaft shows no Golgi outposts localized to the bristle shaft, **E** -~34 h APF, the developing bristle shaft shows no Golgi outpost accumulation, **F**—Developing bristle shaft at -~38 h showing Golgi outposts beginning to accumulate entire bristle shaft. APF—After prepupa formation. Scale bar, 10 μm.

found in cell types other than neurons. In the bristle, Golgi satellites are preferentially localized to the shaft base area. The role of these Golgi satellites remains, however, to be determined. In dendrites, Golgi outposts localize preferentially to branch points, where they are believed to serve as MTOCs (Microtubule organizing centers) [11, 44]. Thus, exact positioning of Golgi outposts is essential for establishing dendritic morphology. Additionally, in mammal neurons, Golgi outposts are more prominent during periods of intensive dendrite growth, while in mature neurons, their frequency decreases with time [14]. This points to not only tight spatial control of Golgi positioning but also to the developmental importance of the process. Here, we focused our analysis on the molecular machinery that regulates organelle positioning, although the function of Golgi satellites in the bristle cell remains to be elucidated.

## The role of MTs in bristle Golgi organization

MTs are the main tracks for long-distance transport in the cell. Thus, MT array integrity and dynamics are essential for proper cellular elongation of both bristles and neurons. In highly polarized neurons, the dendrite and axon differ in terms of their MT array polarity. MTs in the axons of mammal and *Drosophila* neurons are oriented with their plus-ends pointed distally, while *Drosophila* dendrites possess MTs mainly oriented with their minus-ends pointed outwards. This difference in MT orientation is believed to provide directional cues for

differential organelle transport [45–47]. To confirm this hypothesis, we decided to disrupt MT array polarity through expression of mutant Ik2 protein. We found that affecting MT polarity resulted in abnormal Golgi unit localization, together with ectopic trans-Golgi domain appearance. We thus showed that polarized MTs are essential for correct Golgi organization. Specifically, the presence of polarized MTs prevents the appearance of trans-Golgi components within the bristle shaft.

### Possible roles of dynein and kinesin in bristle Golgi organization

Dynein is believed to be the major Golgi architecture effector [5, 48, 49]. In *Drosophila* dendrites, dynein was found to be responsible for dendrite-specific Golgi outpost localization and MT orientation [7, 37, 50]. In neurons, *kinesin-1* plays both indirect and direct role in dendritic selective sorting of Golgi outposts [51]. Indirectly, kinesin depletion alters axon MT orientation, which leads to ectopic Golgi outposts' localization to axons driven by a different motor on misleading MT tracks. Directly, control of *kinesin-1* auto-inhibition prevents entry of Golgi outposts into the axon [51].

We found that in *Dhc64C* mutant flies, bristle shaft Golgi satellites contained an ectopic trans-Golgi compartment that co-localized with cis- and medial-Golgi components. This raises the question of the role of *Dhc64C* in bristle Golgi organization. One explanation is that *Dhc64C* is required for Golgi satellite anchoring/sorting. In this model, dynein may prevent the entry of Golgi units from the soma to the shaft, while in the shaft, dynein specifically anchors the organelle to the lower zone close to soma, preventing the localization of Golgi units to the distant growth area of the cell tip.

A second explanation for the defects in Golgi unit localization observed in *Dhc64C* mutants could be disrupted interactions with the oppositely-directed motor, kinesin. However, taking into consideration MT array organization (i.e., with the minus-end pointed towards the tip) in a fly expressing a dynein mutant, active kinesin would be expected to return Golgi back to the soma. Yet, exactly the opposite is true. Moreover, we demonstrate that, similarly to *Dhc64C* flies, the bristle shaft contained an ectopic trans-Golgi compartment co-localized with cis- and medial Golgi components. In summary, our results shed new light of Golgi organization in polarized cells, such as the *Drosophila* bristle. Still, further study is needed to understand the molecular mechanism by which the combination of a polarized MT network with MT motor protein function affects Golgi bristle organization.

### Supporting information

**S1 Fig. The trans-Golgi marker Gal-T-RFP shows a localization pattern similar to that of GalNacT2-RFP/YFP.** Confocal projections of live developing bristle shafts expressing *Gal-T-RFP* (a trans-Golgi marker): A-B- *Sca-Gal4-UAS-EB1-GFP::UAS-Gal-T-RFP* (WT) developing bristle shaft showing no Gal-T-positive particles at an early stage of bristle development; A—Green EB1-GFP used to highlight the cell perimeter, B- Red—Gal-T-RFP. C-D—Confocal projections of *dynein heavy chain*-mutated background bristle shafts, *Dhc64C^8-1/Dhc64C^4-19* trans-heterozygote bristle shaft expressing *Gal-T-RFP* stained with Oregon-green phalloidin; C–Red- The Gal-T-RFP (trans-Golgi) marker is localized to the entire bristle shaft area, D–Merged—green—phalloidin staining of actin bundles, used here to highlight the cell perimeter, red—Gal-T-RFP. Scale bar, 10 µm.
(EPS)

**S2 Fig. Golgi organization in *Dhc64C* mutant bristle cell soma.** A-C" -Confocal projections of representative of a WT bristle "somal" region from *Dhc64C* mutant background:

A-A"–"Soma" of a *dynein* mutant bristle cell expressing *sca-Gal4::MnII-GFP* (a medial-Golgi marker) stained with anti-dGM130 antibodies (a cis-Golgi marker). A'–Anti-dGM130 antibody staining showing a cis-Golgi compartment localized throughout the bristle soma cytoplasm, A"–Merged image of green *UAS-MnII-GFP* and red anti-dGM130 antibody staining showing co-localization of medial- and cis-Golgi compartments. B-B"–Soma of a *Dhc64C* mutant bristle cell expressing *sca-Gal4::GalNacT2-RFP* (a Trans-Golgi marker), B'- stained with anti-dGM130 antibodies (a cis-Golgi marker), B"–Merged image of GalNacT2-YFP and red anti-dGM130 antibody staining showing co-localization of medial- and Trans-Golgi compartments. C-C"–Soma of a *dynein* mutant bristle cell expressing *sca-Gal4::MnII-GFP* (a medial-Golgi marker) stained with anti-Sec16-antibodies (to identify the ERES): C'–Anti-Sec16 antibody staining showing ERES localized throughout the bristle soma cytoplasm, C"–Merged image of green MnII-GFP and red anti-Sec16 antibody staining showing co-localization of medial-Golgi and ERES components in the bristle cell soma. The scale bar represents 10 μm.
(EPS)

**S3 Fig. Golgi organization in *Lva^{DN}* mutant bristle shaft.** A-D- A Wild type bristle shaft expressing *neur-Gal4-UAS-*GalNacT2-YFP (trans-Golgi marker) stained with anti-dGM130 (cis-Golgi marker) antibodies and phalloidin: A–Gray phalloidin-UV staining of actin bundles in a GalNacT2-YFP expressing bristle, used here to highlight the cell perimeter: B–Green Gal-NacT2-YFP, a trans-Golgi marker, is ectopically localized to the entire bristle shaft area, C–Red anti-dGM130 antibody staining (cis-Golgi marker), D–Merged image of green GalNac-T2-YFP and red anti-dGM130 antibody staining showing co-localization of trans- and cis-Golgi compartments in the mutant bristle shaft. E-H- *neur-UAS-Lava-lamp^{DN}* bristle shaft co-expressing *UAS-GalNacT2-YFP* (trans-Golgi marker) and stained with anti-dGM130 (cis-Golgi marker) antibodies and phalloidin: E–Gray phalloidin-UV staining of actin bundles in a GalNacT2-YFP expressing bristle, used here to highlight the cell perimeter: F–Green GalNac-T2-YFP, a trans-Golgi marker, is ectopically localized to the entire bristle shaft area, G–Red anti-dGM130 antibody staining (cis-Golgi marker), H–Merged image of green GalNacT2-YFP and red anti-dGM130 antibody staining showing co-localization of trans- and cis-Golgi compartments in the mutant bristle shaft. I-L–A wild type *neur-Gal4::UAS-MnII-GFP* bristle shaft expressing stained with anti-Sec16 (ERES) antibodies and phalloidin: I–Gray phalloidin-UV staining of actin bundles in a MnII-GFP-expressing bristle, used here to highlight the cell perimeter: J—Green MnII-GFP, a medial-Golgi marker, is localized to the entire bristle shaft area, K–Red anti-Sec16 antibody staining is localized to the entire shaft area, L–Merged image of green MnII-GFP and red anti-Sec-16-antibody showing co-localization of medial-Golgi and ERES components in the mutant bristle shaft. M-P–*neur-UAS-Lava-lamp^{DN}* bristle shaft co-expressing *UAS-MnII-GFP* stained with anti-Sec16 (ERES) antibodies and phalloidin: M–Gray phalloidin-UV staining of actin bundles in a MnII-GFP-expressing bristle, used here to highlight the cell perimeter: N—Green MnII-GFP, a medial-Golgi marker, is localized to the entire bristle shaft area, O–Red anti-Sec16 antibody staining is localized to the entire shaft area, P–Merged image of green MnII-GFP and red anti-Sec-16-antibody showing co-localization of medial-Golgi and ERES components in the mutant bristle shaft. APF—After prepupa formation. The scale bar represents 10 μm.
(EPS)

**S4 Fig. Golgi organization in *Lva^{DN}* mutant bristle cell soma.** Confocal projections of representative of bristle "somal" region from WT (A-A", C-C", E-E") and *Lva^{DN}* mutant background: A-A"–"Soma" of a WT mutant bristle cell expressing *neur-Gal4::MnII-GFP* (a medial-Golgi marker) stained with anti-dGM130 antibodies (a cis-Golgi marker). A'–Anti-dGM130

antibody staining showing a cis-Golgi compartment localized throughout the bristle soma cytoplasm, A"–Merged image of green MnII-GFP and red anti-dGM130 antibody staining showing co-localization of medial- and cis-Golgi compartments. B-B"–"Soma" of a $Lva^{DN}$ mutant bristle cell expressing *neur-Gal4::MnII-GFP* (a medial-Golgi marker) stained with anti-dGM130 antibodies (a cis-Golgi marker). B'–Anti-dGM130 antibody staining showing a cis-Golgi compartment localized throughout the bristle soma cytoplasm, B"–Merged image of green MnII-GFP and red anti-dGM130 antibody staining showing co-localization of medial- and cis-Golgi compartments. C-C"–Soma of a WT bristle cell expressing *neur-Gal4::GalNac-T2-YFP* (a Trans-Golgi marker), C'- stained with anti-dGM130 antibodies (a cis-Golgi marker), C"–Merged image of GalNacT2-YFP and red anti-dGM130 antibody staining showing co-localization of medial- and Trans-Golgi compartments. D-D"–Soma of a $Lva^{DN}$ mutant bristle cell expressing *GalNacT2-YFP* (a Trans-Golgi marker), D'- stained with anti-dGM130 antibodies (a cis-Golgi marker), D"–Merged image of *UAS-GalNacT2-YFP* and red anti-dGM130 antibody staining showing co-localization of medial- and Trans-Golgi compartments. E-E"–Soma of a WT bristle cell expressing *neur-Gal4::MnII-GFP* (a medial-Golgi marker) stained with anti-Sec16-antibodies (to identify the ERES): E'–Anti-Sec16 antibody staining showing ERES localized throughout the bristle soma cytoplasm, E"–Merged image of green MnII-GFP and red anti-Sec16 antibody staining showing co-localization of medial-Golgi and ERES components in the bristle cell soma. F-F"–Soma of a $Lva^{DN}$ mutant bristle cell expressing MnII-GFP (a medial-Golgi marker) stained with anti-Sec16-antibodies (to identify the ERES): F'–Anti-Sec16 antibody staining showing ERES localized throughout the bristle soma cytoplasm, F"–Merged image of green MnII-GFP and red anti-Sec16 antibody staining showing co-localization of medial-Golgi and ERES components in the bristle cell soma. The scale bar represents 10 μm.
(EPS)

**S5 Fig. Golgi organization in *kinesin heavy chain* mutant bristle cell soma.** A-C" -Confocal projections of representative of bristle "somal" region from *kinesin heavy chain* mutant background: A-A"–"Soma" of a bristle cell co-expressing *sca-Gal4::UAS-MnII-GFP* (a medial-Golgi marker) and *UAS-Khc-RNAi* stained with anti-dGM130 antibodies (a cis-Golgi marker). A'–Anti-dGM130 antibody staining showing a cis-Golgi compartment localized throughout the bristle soma cytoplasm, A"–Merged image of green MnII-GFP and red anti-dGM130 antibody staining showing co-localization of medial- and cis-Golgi compartments. B-B"–Soma of a bristle cell expressing *sca-Gal4::UAS-GalNacT2-YFP* (a Trans-Golgi marker) and *UAS-Khc-RNAi*, B'- stained with anti-dGM130 antibodies (a cis-Golgi marker), B"–Merged image of GalNac-T2-YFP and red anti-dGM130 antibody staining showing co-localization of medial- and Trans-Golgi compartments. C-C"–Soma of a bristle cell expressing *sca-Gal4::UAS-MnII-GFP* (a medial-Golgi marker) and *UAS-Khc-RNAi* stained with anti-Sec16-antibodies (to identify the ERES): C'–Anti-Sec16 antibody staining showing ERES localized throughout the bristle soma cytoplasm, C"–Merged image of green MnII-GFP and red anti-Sec16 antibody staining showing co-localization of medial-Golgi and ERES components in the bristle cell soma. The scale bar represents 10 μm.
(EPS)

**S6 Fig. Golgi organization in $Ik2^{DN}$ bristle cell soma.** A-C" -Confocal projections of representative of bristle "somal" region from $Ik2^{DN}$ mutant background: A-A"–"Soma" of a bristle cell expressing *neur-Gal4::UAS-MnII-GFP* (a medial-Golgi marker) and *UAS- Ik2$^{DN}$* stained with anti-dGM130 antibodies (a cis-Golgi marker). A'–Anti-dGM130 antibody staining showing a cis-Golgi compartment localized throughout the bristle soma cytoplasm, A"–Merged image of green MnII-GFP and red anti-dGM130 antibody staining showing co-localization of

medial- and cis-Golgi compartments. B-B"–Soma of a bristle cell expressing *neur-Gal4::Gal-NacT2-YFP* (a Trans-Golgi marker) and *UAS-Ik2$^{DN}$* , B'- stained with anti-dGM130 antibodies (a cis-Golgi marker), B"–Merged image of GalNacT2-YFP and red anti-dGM130 antibody staining showing co-localization of medial- and Trans-Golgi compartments. C-C"–"Soma" of a bristle cell expressing *neur-Gal4::UAS-MnII-GFP* (a medial-Golgi marker) and *UAS- Ik2$^{DN}$* stained with anti-Sec16-antibodies (to identify the ERES): C'–Anti-Sec16 antibody staining showing ERES localized throughout the bristle soma cytoplasm, C"–Merged image of green MnII-GFP and red anti-Sec16 antibody staining showing co-localization of medial-Golgi and ERES components in the bristle cell soma. The scale bar represents 10 μm.
(EPS)

## Acknowledgments

We thank VDRC Austria and the Bloomington Stock Center for generously providing fly strains. We thank Catherine Rabouille for her enlightening comments and for providing the antibodies against Sec-16.

## Author Contributions

**Conceptualization:** Anna Melkov, Raju Baskar, Uri Abdu.

**Data curation:** Anna Melkov, Raju Baskar.

**Formal analysis:** Anna Melkov, Raju Baskar, Rotem Shachal, Yehonathan Alcalay.

**Methodology:** Anna Melkov, Raju Baskar.

**Project administration:** Anna Melkov, Raju Baskar, Rotem Shachal.

**Supervision:** Uri Abdu.

**Validation:** Anna Melkov, Raju Baskar, Uri Abdu.

**Visualization:** Anna Melkov, Raju Baskar.

**Writing – original draft:** Anna Melkov, Uri Abdu.

**Writing – review & editing:** Anna Melkov, Raju Baskar, Uri Abdu.

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
