## [Decision Letter · Decision Letter 0]

8 Jul 2019

PONE-D-19-15627

The organization of Golgi in Drosophila bristles requires microtubule motor protein function and a properly organized microtubule array

PLOS ONE

Dear Dr. Uri Abdu

Thank you for submitting your manuscript to PLOS ONE. After careful consideration, we feel that it has merit but does not fully meet PLOS ONE’s publication criteria as it currently stands. Therefore, we invite you to submit a revised version of the manuscript that addresses the points raised during the review process.

We would appreciate receiving your revised manuscript by Aug 22 2019 11:59PM. To enhance the reproducibility of your results, we recommend that if applicable you deposit your laboratory protocols in protocols.io, where a protocol can be assigned its own identifier (DOI) such that it can be cited independently in the future. For instructions see: http://journals.plos.org/plosone/s/submission-guidelines#loc-laboratory-protocols

We look forward to receiving your revised manuscript.

Kind regards,

Maria Grazia Giansanti

Academic Editor

PLOS ONE

**Journal Requirements:**

**Comments to the Author**

1. Is the manuscript technically sound, and do the data support the conclusions?

Reviewer #1: Partly

Reviewer #2: Partly

2. Has the statistical analysis been performed appropriately and rigorously? 

Reviewer #1: Yes

Reviewer #2: No

3. Have the authors made all data underlying the findings in their manuscript fully available?

Reviewer #1: Yes

Reviewer #2: Yes

4. Is the manuscript presented in an intelligible fashion and written in standard English?

Reviewer #1: Yes

Reviewer #2: Yes

5. Review Comments to the Author

Reviewer #1: In their manuscript Abdu and colleagues investigate the organization of Golgi stacks in Drosophila bristles. Similar to dendrites, the bristles of the fly contain Golgi outposts, which are likely used to supply processed secretory/membrane proteins at sites far from the cell's soma. This is an interesting model system, and the authors have done a nice work to characterize the microtubule dependent formation of these Golgi outposts. I believe this will be a good contribution to the literature, but suggest that the authors consider some revision as detailed below.

1. It would be good to document the number of repeats for each measurement where statistics was used (displayed in table 1).

2. The dual color staining of the Golgi in different combinations does not give this reviewer a lot of confidence that the Golgi mini-stacks are properly organized. Three or even four color images should be possible where secondary antibodies are used (e.g. for dGM130 antibody staining), showing that the Golgi (and ERES) are in fact properly polarized in a cis-trans direction.

3. The use of the IK2 dominant negative reagent is very interesting, but I had to dig deep and hard to find information on what this reagent (or for that matter the IK2 protein in this context) does. Some more information would be nice, this could be in the introduction, the results or even in the discussion sections, depending on the authors' preference, but I feel it would make the paper's message easier to understand.

4. The authors' use of the term "mini-stacks" is not correct in my opinion. Mini-stack is generally used to mean Golgi stacks that are not assembled in a ribbon, and are as such commonly used by mammalian cell biologists (although not exclusive to the mammalian system of course). Here the authors use it seemingly to mean stacks lacking a cisterna (in this case the trans cisterna). I would caution against that, and would rather use a different term.

Reviewer #2: In this manuscript, Melkov and colleagues present a series of experiments aimed to describe the organization of Golgi in highly elongated bristles upon disruption of microtubule motor protein function in Drosophila melanogaster.

The authors used fluorescently labeled transgenic lines to visualize the medial and trans-Golgi compartments, and Sec16 and GM130 antibodies to show the distribution of ERES and cis-Golgi in WT and different mutant backgrounds. They measured the distribution and composition of Golgi in the bristle shaft tip and the base region and found complete Golgi in the entire bristle shaft when they interfere with microtubule polarity.

Overall, the experimental results are interesting, but the experiments have several fundamental problems and the interpretation of the results is not satisfactory. Several questions need to be answered in order to be published in PlOS One.

Major comments:

1. To give a more definite answer regarding the involvement of microtubule polarity in the elongated bristle shaft cell the whole cell should be tested. They compare only the bristle shaft tip and base regions, but not the middle region or the soma region. How is the Golgi distribution in the soma in the different mutants? Is it possible, that the abnormal Golgi distribution is only the indirect consequences of the general problem in cell homeostasis and globally disturbed Golgi organization?

2. They used the UAS/Gal4 system to drive transgenes in the bristle. While ManII-GFP signal is strong in Fig 1A and C, it is weaker when it is co-expressed by the GalNacT2-RFP, probably due to the fact that two transgenes use the Gal4 source in those flies. It is not clear which driver did they use, the neur-Gal4 or the sca-Gal4 driver in the WT. Why do they used the Sca-Gal4 on Khc RNAi and Dhc64 mutant, but Neur-Gal4 on the Lava-lamp dominant negative background?

On the same line, they used different driver lines in the WT and the mutants when they visualized the developmental distribution of Golgi outpost in different pupal stages (Fig6). This is especially important when the authors compare statistically the Golgi localization (particle area, density) in WT and mutants (Table1).

3. Based on the fact that Golgi distribution changes during pupal development, it will be important to mention what is the age of the pupae of the WT and mutants on Fig1-5.

4. Ik2 dominant negative mutant bristles are shorter than WT, therefore the statistical analysis could be misleading when there is a much shorter bristle.

5. On page 20 they state that ~32h APF complete Golgi units could be detected, but only MnII-GFP is presented, no cis- or trans-Golgi marker is shown.

6. The 3 and 4 pupae per genotype is not enough to do proper statistical analyses. It is not clear whether they tested 4, 5 or 6 bristles altogether or per one pupae in each genotype. Especially in the case of Khc-RNAi, I suggest doubling the tested bristles.

7. Why do they used Dhc64C8-1/Dhc64C902 mutant combination to visualize the cis-Golgi and ERES distribution (Fig2A-C and G-I), but Dhc64C8-1/Dhc64C4-19 combination to test the trans-Golgi GalNacT2-YFP (Fig2D-F)?

8. It is shown the actin distribution in WT and Khc-RNAi, but not in Dhc mutant, LvaDN or Ik2DN mutants. It will be necessary to include those images too.

9. Fig3 C shows only the MnII-GFP localization, but not the cis- or trans-Golgi marker. It will be useful to include those as well.

10. Table 1 is including data from the MnII-GFP measurement, but it will be useful to include the cis- or trans-Golgi data also. Moreover, the precise description (measurement based on MnII-GFP) of Table 1 is missing.

11. There are several generalizations in the text. Organization of the Golgi is specialized in Drosophila, therefore the authors should be careful when they generalize about Golgi organization. (“kinesin is present in the Golgi membranes” (Ref. 38). There is not much information about kinesin function in Golgi trafficking, therefore, the Golgi distribution in the soma of the mutant should be included.

12. On page 22 they referred, that in the soma of the larval neurons contains the ring-shaped complete Golgi, however, it was shown that during spermatogenesis, the Golgi based acroblast organization is similar. (ref: Farkas RM. et. al. Molecular Biology of the Cell (2003) Vol.14, 190–200, Fari K et. al. Biology Open (2016) 5, 1102-1110)

In addition, the manuscript would benefit from careful editing by a native English speaker; there are missing words, sentences where the subject and verb do not agree, places where there are extra or missing commas, sentences where the word order is confusing or spelling mistakes (mammal neurons instead of mammalian neurons, etc.).

Minor comments:

1. Description of Fig4 has several confusing sentences.

2. Scale bar on Fig3 B, B’ is not correct.

3. Dhc4-19 allele is missing from the stock list of Materials and methods.

4. On Page 20. they use Khc64C, which should be Dhc64C.

5. In the “Kinesin heavy chain section”, Dhc64 is mentioned in the 7th line, but that one is the mutant, not the gene, therefore it is should be labeled correctly.

6. PLOS authors have the option to publish the peer review history of their article (what does this mean?). If published, this will include your full peer review and any attached files.

Reviewer #1: Yes: Daniel Ungar

Reviewer #2: No

---

## [Author Response · Author response to Decision Letter 0]

4 Sep 2019

Reviewer #1: In their manuscript Abdu and colleagues investigate the organization of Golgi stacks in Drosophila bristles. Similar to dendrites, the bristles of the fly contain Golgi outposts, which are likely used to supply processed secretory/membrane proteins at sites far from the cell's soma. This is an interesting model system, and the authors have done a nice work to characterize the microtubule dependent formation of these Golgi outposts. I believe this will be a good contribution to the literature, but suggest that the authors consider some revision as detailed below.

1. It would be good to document the number of repeats for each measurement where statistics was used (displayed in table 1).

We agree with this comment and have added the number of repeats for each experiment throughout the revised text.

2. The dual color staining of the Golgi in different combinations does not give this reviewer a lot of confidence that the Golgi mini-stacks are properly organized. Three or even four color images should be possible where secondary antibodies are used (e.g. for dGM130 antibody staining), showing that the Golgi (and ERES) are in fact properly polarized in a cis-trans direction.

In this paper, we used the different Golgi marker only for marking the different Golgi compartments and not for revealing the spatial organization of the Golgi compartments in relation to each other. Indeed, for such analysis more experiments and tools are needed, which we emphasized was not the aim in our paper. 

3. The use of the IK2 dominant negative reagent is very interesting, but I had to dig deep and hard to find information on what this reagent (or for that matter the IK2 protein in this context) does. Some more information would be nice, this could be in the introduction, the results or even in the discussion sections, depending on the authors' preference, but I feel it would make the paper's message easier to understand.

We agree with this comment and have added the following description to the Results: ”Ik2DN is a defective kinase where lysine at position 41 was changed to alanine (Ik2K41A)”.

4. The authors' use of the term "mini-stacks" is not correct in my opinion. Mini-stack is generally used to mean Golgi stacks that are not assembled in a ribbon, and are as such commonly used by mammalian cell biologists (although not exclusive to the mammalian system of course). Here the authors use it seemingly to mean stacks lacking a cisterna (in this case the trans cisterna). I would caution against that, and would rather use a different term.

We agree with this comment, and indeed the bristle shaft contains Golgi stacks that lack trans cisterna and thus could not be termed mini-stacks. Thus, we decided to call this unique structure incomplete Golgi. 

Reviewer #2: In this manuscript, Melkov and colleagues present a series of experiments aimed to describe the organization of Golgi in highly elongated bristles upon disruption of microtubule motor protein function in Drosophila melanogaster.

The authors used fluorescently labeled transgenic lines to visualize the medial and trans-Golgi compartments, and Sec16 and GM130 antibodies to show the distribution of ERES and cis-Golgi in WT and different mutant backgrounds. They measured the distribution and composition of Golgi in the bristle shaft tip and the base region and found complete Golgi in the entire bristle shaft when they interfere with microtubule polarity.

Overall, the experimental results are interesting, but the experiments have several fundamental problems and the interpretation of the results is not satisfactory. Several questions need to be answered in order to be published in PlOS One.

Major comments:

1. To give a more definite answer regarding the involvement of microtubule polarity in the elongated bristle shaft cell the whole cell should be tested. They compare only the bristle shaft tip and base regions, but not the middle region or the soma region. How is the Golgi distribution in the soma in the different mutants? Is it possible, that the abnormal Golgi distribution is only the indirect consequences of the general problem in cell homeostasis and globally disturbed Golgi organization?

We agree with the comment referring to revealing the localization of Golgi in the somatic region of WT, as compared to the mutants. We analyzed the localization pattern of the Golgi in the somal region of the wild type and all other mutant backgrounds. New figures were generated and since no obvious changes in the localization pattern of the Golgi were noted, we added these as Supplemental figures. 

As to the localization pattern at the bristle shaft, as explained in the Materials and methods, we divided the entire bristle shaft into two parts, thus there is no need for a middle part. 

(Page 7-“A Z-stack projection of the bristle shaft was divided into two halves exactly in the middle of the shaft length, resulting in two sections of even length, with one part being close to the base (soma) and the other being distal to the base, referred to as the tip”).

2. They used the UAS/Gal4 system to drive transgenes in the bristle. While ManII-GFP signal is strong in Fig 1A and C, it is weaker when it is co-expressed by the GalNacT2-RFP, probably due to the fact that two transgenes use the Gal4 source in those flies. It is not clear which driver did they use, the neur-Gal4 or the sca-Gal4 driver in the WT. 

We agree with this comment and have added the Gal4 used in this figure. 

Why do they used the Sca-Gal4 on Khc RNAi and Dhc64 mutant, but Neur-Gal4 on the Lava-lamp dominant negative background? On the same line, they used different driver lines in the WT and the mutants when they visualized the developmental distribution of Golgi outpost in different pupal stages (Fig6). This is especially important when the authors compare statistically the Golgi localization (particle area, density) in WT and mutants (Table1).

In this study, we alternated between the different Gal-4 lines due to technical problems. Both over-expression of Lava-lampDN and Ik2DN with sca-Gal4 was not as effective as with Neur-Gal4. We agree with the reviewer’s comment that it is difficult to compare statistically, so we omitted the statistical analysis between Ik2DN and Dhc and Khc, and just describe the effect of mutations of Ik2 on Golgi organization. Thus. the following text was added: “First, we found that expression with neur-Gal4 and not Sca-Gal4 generated typical Ik2 bristle defects. Thus, we used this Gal4 for further examinations” . Later in the text, we added: “Since we used different Gal4s for Golgi expression in Dhc64C and Khc-RNAi (Sca-Gal4), in comparison to Ik2DN expression (neur-Gal4), we were not able to statistically compare Golgi parameter (area and density) between them.” 

3. Based on the fact that Golgi distribution changes during pupal development, it will be important to mention what is the age of the pupae of the WT and mutants on Fig1-5.

We agree with this comment and have added the age of the pupa in all figures. 

4. Ik2 dominant negative mutant bristles are shorter than WT, therefore the statistical analysis could be misleading when there is a much shorter bristle.

Based on comment 2 of reviewer #2, we decided that we could not preform statistical analysis between Ik2 and the rest of the mutants. As such, this comment is not relevant. 

5. On page 20 they state that ~32h APF complete Golgi units could be detected, but only MnII-GFP is presented, no cis- or trans-Golgi marker is shown.

We agree with this comment and have changed the text throughout figure 6 caption accordingly. 

6. The 3 and 4 pupae per genotype is not enough to do proper statistical analyses. It is not clear whether they tested 4, 5 or 6 bristles altogether or per one pupae in each genotype. Especially in the case of Khc-RNAi, I suggest doubling the tested bristles.

We agree with this comment and have doubled our tested bristle analysis. Now, we have at least 5 to 6 pupa and at least 3 bristles per pupa. 

7. Why do they used Dhc64C8-1/Dhc64C902 mutant combination to visualize the cis-Golgi and ERES distribution (Fig2A-C and G-I), but Dhc64C8-1/Dhc64C4-19 combination to test the trans-Golgi GalNacT2-YFP (Fig2D-F)?

There is no specific reason why we used these two different alleles since we (Melkov et al., 2016) and others (Gepner et al., 1996 and Li et al., 2008) previously showed that both alleles, Dhc64C902 and Dhc64C4-19, are strong amorphic alleles which give the exact bristle phenotypes in combinations with Dhc64C8-1

8. It is shown the actin distribution in WT and Khc-RNAi, but not in Dhc mutant, LvaDN or Ik2DN mutants. It will be necessary to include those images too.

We agree with this comment and have performed this analysis again and include actin staining. 

9. Fig3 C shows only the MnII-GFP localization, but not the cis- or trans-Golgi marker. It will be useful to include those as well.

We agree with this comment and have performed this analysis. The new figure 3 now includes other Golgi domains markers. 

10. Table 1 is including data from the MnII-GFP measurement, but it will be useful to include the cis- or trans-Golgi data also. Moreover, the precise description (measurement based on MnII-GFP) of Table 1 is missing.

We agree with this comment and have added a new data set for GalNacT2-YFP as well; all of these new results are summarized in Table 2. 

11. There are several generalizations in the text. Organization of the Golgi is specialized in Drosophila, therefore the authors should be careful when they generalize about Golgi organization. (“kinesin is present in the Golgi membranes” (Ref. 38). There is not much information about kinesin function in Golgi trafficking, therefore, the Golgi distribution in the soma of the mutant should be included.

We agree with this comment and have changed the text accordingly. Also, as mentioned in comment #1, Golgi distribution in the soma of the mutant was also included.

12. On page 22 they referred, that in the soma of the larval neurons contains the ring-shaped complete Golgi, however, it was shown that during spermatogenesis, the Golgi based acroblast organization is similar. (ref: Farkas RM. et. al. Molecular Biology of the Cell (2003) Vol.14, 190–200, Fari K et. al. Biology Open (2016) 5, 1102-1110)

We agree with this comment and have added Golgi-based acroblast organization in sperrmatogeneis to the text. The following text and references were added to the discussion: ” “During spermatogenesis the Golgi-based acroblast has a ring shape.

In addition, the manuscript would benefit from careful editing by a native English speaker; there are missing words, sentences where the subject and verb do not agree, places where there are extra or missing commas, sentences where the word order is confusing or spelling mistakes (mammal neurons instead of mammalian neurons, etc.).

The paper was edited by a native English editor.

Minor comments:

1. Description of Fig4 has several confusing sentences.

We have changed the text

2. Scale bar on Fig3 B, B’ is not correct.

The scale bar was fixed.

3. Dhc4-19 allele is missing from the stock list of Materials and methods.

We have added this allele to the stock list of Materials and methods.

4. On Page 20. they use Khc64C, which should be Dhc64C.

We changed the text accordingly. 

5. In the “Kinesin heavy chain section”, Dhc64 is mentioned in the 7th line, but that one is the mutant, not the gene, therefore it is should be labeled correctly.

We changed the text accordingly.

---

## [Decision Letter · Decision Letter 1]

17 Sep 2019

The organization of Golgi in Drosophila bristles requires microtubule motor protein function and a properly organized microtubule array

PONE-D-19-15627R1

Dear Dr. Abdu,

We are pleased to inform you that your manuscript has been judged scientifically suitable for publication and will be formally accepted for publication once it complies with all outstanding technical requirements.

With kind regards,

Maria Grazia Giansanti

Academic Editor

PLOS ONE

Additional Editor Comments (optional):

Reviewers' comments:

Reviewer's Responses to Questions

**Comments to the Author**

1. If the authors have adequately addressed your comments raised in a previous round of review and you feel that this manuscript is now acceptable for publication, you may indicate that here to bypass the “Comments to the Author” section, enter your conflict of interest statement in the “Confidential to Editor” section, and submit your "Accept" recommendation.

Reviewer #1: All comments have been addressed

Reviewer #2: All comments have been addressed

2. Is the manuscript technically sound, and do the data support the conclusions?

Reviewer #1: Yes

Reviewer #2: Yes

3. Has the statistical analysis been performed appropriately and rigorously? 

Reviewer #1: Yes

Reviewer #2: Yes

4. Have the authors made all data underlying the findings in their manuscript fully available?

Reviewer #1: Yes

Reviewer #2: Yes

5. Is the manuscript presented in an intelligible fashion and written in standard English?

Reviewer #1: Yes

Reviewer #2: Yes

6. Review Comments to the Author

Reviewer #1: (No Response)

Reviewer #2: The authors have corrected the mistakes from the original version and included the requested images. They also provided more statistic data for the phenotypic defect of different mutants, therefore the conclusions are convincing in this version of the manuscript.

7. PLOS authors have the option to publish the peer review history of their article (what does this mean?). If published, this will include your full peer review and any attached files.

Reviewer #1: No

Reviewer #2: No

---

## [Editor Report · Acceptance letter]

23 Sep 2019

PONE-D-19-15627R1 

The organization of Golgi in *Drosophila* bristles requires microtubule motor protein function and a properly organized microtubule array 

Dear Dr. Abdu:

I am pleased to inform you that your manuscript has been deemed suitable for publication in PLOS ONE. Congratulations! Your manuscript is now with our production department. 

With kind regards,

on behalf of

Dr. Maria Grazia Giansanti 

Academic Editor

PLOS ONE